# Multiparity and Aging Impact Chondrogenic and Osteogenic Potential at Symphyseal Enthesis: New Insights into Interpubic Joint Remodeling

**DOI:** 10.3390/ijms24054573

**Published:** 2023-02-26

**Authors:** Lizandra Maia de Sousa, Bianca Gazieri Castelucci, Paula Andrea Saenz Suarez, Ingrid Iara Damas, Fernanda Viviane Mariano, Paulo Pinto Joazeiro, Sílvio Roberto Consonni

**Affiliations:** 1Laboratory of Cytochemistry and Immunocytochemistry, Department of Biochemistry and Tissue Biology, Institute of Biology (IB), State University of Campinas (Unicamp), Campinas 13083-970, Brazil; 2Laboratory of Immunometabolism, Department of Genetic, Evolution, Microbiology and Immunology, Institute of Biology (IB), State University of Campinas (Unicamp), Campinas 13083-970, Brazil; 3Laboratory of Molecular Pathology, Department of Pathology, Division of Pathological Anatomy, School of Medical Sciences (FCM), State University of Campinas (Unicamp), Campinas 13083-970, Brazil

**Keywords:** parity, aging, cartilage, pubic symphysis, pregnancy, chondrogenesis

## Abstract

Pregnancy and childbirth cause adaptations to the birth canal to allow for delivery and fast recovery. To accommodate delivery through the birth canal, the pubic symphysis undergoes changes that lead to the interpubic ligament (IpL) and enthesis formation in primiparous mice. However, successive deliveries influence joint recovery. We aimed to understand tissue morphology and chondrogenic and osteogenic potential at symphyseal enthesis during pregnancy and postpartum in primiparous and multiparous senescent female mice. Morphological and molecular differences were found at the symphyseal enthesis among the study groups. Despite the apparent incapacity to restore cartilage in multiparous senescent animals, the symphyseal enthesis cells are active. However, these cells have reduced expression of chondrogenic and osteogenic markers and are immersed in densely packed collagen fibers contiguous to the persistent IpL. These findings may indicate alterations of key molecules in the progenitor cell population maintenance of the chondrocytic and osteogenic lineages at the symphyseal enthesis in multiparous senescent animals, possibly compromising the mouse joint histoarchitecture recovery. This sheds light on the distention of the birth canal and the pelvic floor that may play a role in pubic symphysis diastasis (PSD) and pelvic organ prolapse (POP), both in orthopedic and urogynecological practice in women.

## 1. Introduction

The integrity of the ligaments maintains physiological pelvic support. The pelvic bone and the soft tissues surrounding it compose the birth canal in humans and animal models, and the bony part contains the pubic symphysis [1,2]. Pregnancy and childbirth impose drastic adaptations to the birth canal to allow for delivery and fast recovery to a similar pre-gestational state [3,4,5,6]. However, epidemiological studies suggest that many women fail to fully recover from this event [7,8], and the excessive widening of the pubic symphysis may result in pubic symphysis diastasis (PSD) [9]. As the birth canal is directly related to pelvic organ support [10,11], damage to any component of this canal may result in a loss of support and, consequently, pelvic organ prolapse (POP) in women. Childbirth is the most significant independent risk factor for developing pelvic floor dysfunction [12,13,14], and most multiparous women have some anatomical evidence of pelvic floor dysfunction [15,16,17].

POP can occur at any age; however, older women over the age of 50 have a higher prevalence across the world [18]. Moreover, aging is one of the factors for the lesser adaptation of tissues to their functions [19]. However, up to now, the mechanisms by which pregnancy, childbirth, and aging lead to the weakening of pelvic organs and supporting structures of these organs are not known [20,21,22].

Similar to that observed in humans, experimental deletions of key molecules in elastin fibrillogenesis can induce POP in mice [23,24,25,26]. The observation of POP in the reproductive matrix of Fibulin 5 knockout mice showed that these animals progressed to severe POP as a function of age and parity [27]. Like humans, increasing parity and age appear to accelerate the disease process in murine models. In addition, retired breeders demonstrated signs of reproductive senescence (approximately eight months of age) after six to eight consecutive deliveries with declining fertility [28]. Therefore, mouse reproductive matrices seem to be models for evaluating the biomechanical properties of the birth canal.

The mouse pubic symphysis (PS) is a fibrocartilaginous joint that makes up part of the musculoskeletal system of the birth canal, which supports the pelvic organs in providing stability and helping to neutralize the compression forces acting on the pelvis during movement [4,29,30]. The joint undergoes drastic changes that lead to the formation of an interpubic ligament (IpL) to accommodate the birth canal [1,30,31]. Cell cytoskeleton modification and elastogenesis are necessary for the facilitated occurrence of labor and its success in mice [32,33]. During IpL formation, in primiparous mice, a transitional structure with gradations and differential organization of cells and extracellular matrix (ECM) components, called enthesis, develops and connects the IpL to the bones of the pelvis [34].

Some biological factors are necessary for the development, homeostasis, and remodeling of mouse entheses, such as the expression of Runt-related transcription factor 2 (*Runx-2*), SRY-Box transcription factor 9 (*Sox-9*), and Collagen type II alpha 1 chain (*Col2a1*) genes [35]. The spatiotemporal expression of these genes is very similar to that found in the balance of chondrogenesis and osteogenesis, as observed in endochondral ossification in in vitro human and animal models in vitro and in vivo (animal models and human) [36,37,38,39,40]. Therefore, the investigation of gene expression and immunolocalization of Sox-9, Runx-2, and Col2a1 may help to evaluate the homeostatic balance of the enthesis remodeling in the mouse pubic symphysis, since they are classically involved in the osteogenesis and chondrogenesis processes. After the first delivery, the fibrocartilaginous tissue is restored, and the mouse interpubic joint quickly closes, approaching the dimensions of a non-pregnant animal [33,41], probably due to the time expression-dependent chondrocytic lineage markers for successful recovery of fibrocartilaginous tissue [34]. On the other hand, senescent multiparous mice present cellular and extracellular characteristics of the interpubic ligament related to a possible loss of tissue return capacity in the postpartum period [33]. Thus, considering the ligament remodeling biology and its contribution to connective tissue disorders, as well as the animal model of reproductive matrices, in this work, we aimed to understand the effects of parity and age on chondrogenesis and osteogenesis at the symphyseal enthesis in primiparous mice and multiparous senescent mice during pregnancy and postpartum.

## 2. Results

### 2.1. Tissue Morphology and Cell Phenotype of Mouse Symphyseal Enthesis Remodeling during Pregnancy and Postpartum

To characterize the mouse symphyseal enthesis, we realized a systematic analysis of tissue morphology and cell phenotype under light and transmission electron microscopies. In primiparous female mice, our results evidenced fibrochondrocytes in the fibrocartilaginous extracellular matrix (Figure 1A and Figure 2A) in the middle of the transversal sections of the pubic joint. Fibrochondrocytes mostly form the cell population in isogenous groups. The ultrastructure shows abundant rough endoplasmic reticulum in the cytoplasm and loose chromatin in the nucleus (Figure 1E). On both sides of the fibrocartilage, there is a basophilic pad of hyaline cartilage covering the pubic bones (PB) (Figure 2A). Under polarized light, the yellow and orange birefringent collagen fibers around the fibrochondrocytes and the perpendicular arrangement of the fibers to the joint opening allow us to evidence the collagen fiber organization and orientation (Figure 2A).

At late pregnancy in primiparous female mice, the typical decrease in basophilia at the hyaline cartilage ECM and the development of fibrocartilaginous enthesis allow us to distinguish two distinct regions: the bone proximal region (BPR) and the bone distal region (BDR). The BPR is next to the PB and is composed of numerous rounded cells close to each other and loose chromatin (Figure 1B). The BDR connects the BPR and IpL and is composed of elongated cells and a greater amount of ECM. The BDR cells are not parallelly aligned to the interpubic joint opening, as seen in the fibroblast-like cells in the IpL (Figure 1B).

The ultrastructure of both regions of the symphyseal enthesis shows cells with abundant rough endoplasmic reticulum in their cytoplasm and loose chromatin in their nucleus (Figure 1F). In addition, the ECM at D18 shows a gradual transition in the organization of collagen fibers between enthesis and IpL with the presence of crimps (Figure 2B).

After delivery, the postpartum remodeling in primiparous female mice leads to the IpL reabsorption process, as continuously observed in 3dpp (Figure 1C). The BPR shows less ECM between numerous cells with elongated morphologies and an orientation perpendicular to the joint opening. Further, cells show a reorganization into isogenous groups (Figure 1C) and an elongated morphology with a well-developed rough endoplasmic reticulum and loose chromatin (Figure 1G). Similar to D18, ECM at 3dpp also shows a gradual transition in the organization of collagen fibers between enthesis and IpL. Though the BPR demonstrates green birefringence of collagen fiber, the BDR shows a yellowish/orange color close to the IpL (Figure 2C). The crimps are present in both the BPR and the BDR (arrows and inset in Figure 2C).

At 21dpp, there is significant IpL reabsorption to restore the histoarchitecture in primiparous female mice (Figure 1D). The BPR and BDR of the enthesis are almost indistinguishable, giving rise to hyaline-like cartilage and fibrocartilage-like tissues observed in D12 primiparous female mice. Next to the PB, isogenous groups are immersed in the ECM with a glass-like appearance (Figure 1D) and a noticeable basophilic territorial matrix (arrow in Figure 2D). The cells show active metabolic characteristics such as a well-developed rough endoplasmic reticulum and loose chromatin (Figure 1H). The collagen fibers show orange birefringence around fibrochondrocyte-like cells (see inset Figure 2D), which is also the case in D12 primiparous female mice.

In contrast, D12 multiparous senescent female mice show different interpubic joint histoarchitecture during their sixth pregnancy compared to the primiparous group. Despite the BPR and the BDR being indistinguishable, the symphyseal enthesis remains attached between the PB and IpL after successive pregnancies (Figure 1I). The cells near the PB show a distinguished cytoplasm with cellular projections, immersed in densely packed collagen fibers (Figure 1I), elongated features with a slightly condensed chromatin (Figure 1M). The ECM presents birefringent collagen fibers ranging from green to yellow that are organized mostly parallel to the joint opening (Figure 2E). The birefringence denotes a gradual transition in the organization of collagen fibers between symphyseal enthesis and IpL.

Next, in D18 multiparous senescent female mice, the distinction of the symphyseal enthesis regions seems to still be unclear, and the cells show elongated morphologies immersed in the fibrillar ECM (Figure 1J). At enthesis, the elongated cells show characteristics of active cells, such as a well-developed rough endoplasmic reticulum and loose chromatin (Figure 1N), similar to D18 in primiparous female mice. The ECM is predominantly comprised of orange birefringent and collagen fibers aligned parallelly to the joint opening with crimps close to the IpL (Figure 2F).

After delivery, the symphyseal enthesis in multiparous female mice shows numerous cells with fusiform morphologies, loose chromatin, and a poorly developed rough endoplasmic reticulum at 3dpp (Figure 1K,O). In addition, the collagen fiber birefringence displays predominantly green with some yellowish regions, where crimps can be observed (arrows and inset in Figure 2G).

In multiparous female mice, there seems not to be a fully cartilaginous restoration after the sixth delivery at the interpubic joint. The symphyseal enthesis comprises various cells with abundant or scarce cytoplasm, both immersed in densely packed collagen fibers (Figure 1L). Some cells with sparse cytoplasm show few organelles and condensed chromatin mainly associated with the nuclear envelope (Figure 1P). The collagen fiber birefringence varies between green and yellowish, and crimps can be found at the symphyseal enthesis (arrow and inset in Figure 2H).

### 2.2. Gene Expression of Chondrogenic and Osteogenic Transcription Factors at the Mouse Interpubic Joint Remodeling during Pregnancy and Postpartum

To exploit the spatiotemporal expression of chondrogenic and osteogenic transcription factors at the interpubic joint remodeling during pregnancy and postpartum in primiparous and multiparous senescent female mice, we have analyzed *Col2a1*, *Runx-2*, and *Sox-9* gene expression by real-time PCR (qPCR). In primiparous female mice, the *Col2a1* gene expression pattern reveals significant downregulation at D18 compared to the control group (D12); however, after birth, the *Col2a1* gene expression level seems to be reestablished similarly to D12. In multiparous senescent female mice, there is an overall reduction in *Col2a1* gene expression, except at 3dpp, when compared to D12 and D18. Comparisons of the same day of study between multiparous senescent and primiparous mice demonstrate a significant reduction in *Col2a1* gene expression at D12 and 21dpp in multiparous female mice (Figure 3A).

In primiparous female mice, the *Runx-2* gene expression pattern shows significant upregulation at 3dpp when compared to D12 and D18 and at 21dpp when compared to D12. In contrast, the *Runx-2* gene expression is statistically similar among all days of study in multiparous senescent mice. Comparisons of the same day of study between multiparous senescent and primiparous mice show a significant reduction in *Runx-2* in 3dpp and 21dpp in multiparous senescent female mice (Figure 3B).

The *Sox-9* gene expression in primiparous female mice reveals significant upregulation at late pregnancy (D18) and early postpartum (3dpp); however, at 21dpp, *Sox-9* gene expression seems to return to a similar level at D12, with a significant downregulation when compared to 3dpp. In multiparous senescent female mice, *Sox-9* gene expression does not change significantly during pregnancy. However, during postpartum, a significant increase occurs at 3dpp compared to D12 and D18, and a significant reduction occurs at 21dpp compared to 3dpp. Comparisons of the same day of study between multiparous senescent and primiparous mice show a significant reduction in *Sox-9* at D18 in multiparous senescent female mice (Figure 3C).

### 2.3. Immunolocalization of Chondrogenic and Osteogenic Markers at the Mouse Symphyseal Enthesis Remodeling during Pregnancy and Postpartum

The immunolocalization of the Runx-2 and Sox-9 proteins at the symphyseal enthesis shows a differential spatiotemporally pattern between the primiparous and multiparous senescent female mice. In primiparous female mice, at D12, the hyaline cartilage and the fibrocartilage comprise about 81% of immunopositive cells for Runx-2 and 65% for Sox-9 (Figure 4A and Figure 5A). At D18, the highest chromogen intensity, the percentage of immunopositive cells is about 82% for Runx-2 and 71% for Sox-9 at the BPR and BDR (Figure 4B and Figure 5B). After delivery, there is a decrease in the chromogen intensity of the BPR and BDR at 3dpp, and nearly 78% of cells are immunopositive for both Runx-2 and Sox-9 (Figure 4C and Figure 5C). At 21dpp, about 67% of cells are immunopositive for Runx-2, and 61% for Sox-9 (Figure 4D and Figure 5D). The immunostaining for Runx-2 and Sox-9 is evident in chondrocytes at the restored interpubic hyaline cartilage joint at 21dpp.

At the symphyseal enthesis in multiparous senescent female mice, Runx-2 and Sox-9 proteins are found in about 64% and 26% of cells, respectively, at D12 (Figure 4E and Figure 5E). At D18, Runx-2 and Sox-9 proteins are found in nearly 44% and 26% of cells, respectively (Figure 4F and Figure 5F). Following delivery, at 3dpp and 21dpp, there are about 76% and 62% immunopositive cells for Runx-2, respectively; and about 53% and 33% for Sox-9, respectively, at symphyseal enthesis (Figure 4G,H and Figure 5G,H).

## 3. Discussion

The mechanisms by which pregnancy, childbirth, and aging lead to weakened pelvic organs and supporting structures in women are still unknown [20,21,22]. Thus, considering the connective tissue disorders in women, such as PSD and POP; the prevalence of POP associated with the number of births and age; as well as the animal model of reproductive matrices for evaluating the effects of connective tissue remodeling on the biomechanical properties of the birth canal, our work has evidenced multiparity and aging as important factors that impact chondrogenic and osteogenic potential at mouse symphyseal enthesis. This evidence was confirmed through cell phenotype, ECM arrangement, gene expression, and immunolocalization of Sox-9, Runx-2, and Col2a1 analyses during mouse interpubic joint remodeling. Although the mechanism may not be the same as the natural progression and development of connective tissue disorders in women, the rodent may provide important information on tissue remodeling, which may contribute to investigations associated with orthopedic and urogynecological practice in women [2].

The previous work of our research group has demonstrated that the morphological changes mediated by the physiological process of pregnancy in primiparous animals led to the development of a structure with ECM gradations and a distinct cell population called enthesis, which is present from the end of the first pregnancy to the first day postpartum [34]. However, during the sixth pregnancy of senescent multiparous animals, we have shown that the symphyseal enthesis is present at D12 and apparently does not display significant morphological changes in its structure at D18, 3dpp, and 21dpp. In addition, the evaluation of chondrogenic and osteogenic factors in the interpubic joint highlighted a fine spatiotemporal regulation of these determining factors in cell differentiation at the interpubic joint remodeling.

In primiparous animals, throughout the process of IpL formation and retraction of the hyaline cartilage pads at the end of pregnancy, two regions of the symphyseal enthesis are distinguished: the BPR and the BDR [34]. At the end of the first pregnancy, on D19, the fibrocartilage transition zone at the bone interface is a fibrocartilaginous enthesis [34,42]. However, at postpartum, the symphyseal enthesis was characterized as fibrous due to the anchoring of collagen fibers directly to the bone, allowing for direct interaction between the PB and the IpL [34,43].

Unlike in primiparous females, in multiparous senescent females, during the sixth pregnancy and postpartum, a structure similar to the fibrous enthesis was observed on all days of our study, in which the collagen fibers of the IpL were directly continuous with the PB. Fibrous entheses are subject to the highest risk of rupture, since the absence of the fibrocartilaginous portion in this type of enthesis directly alters its biomechanical characteristics in animal models [44,45,46,47,48]. Furthermore, epidemiological studies have suggested that many women fail to recover from severe distention of the birth canal caused by the physiological processes of pregnancy, which may play an important role in causing POP [7,8]. Thus, symphyseal enthesis structure is morphologically altered, and, possibly, its biomechanics may be also altered in multiparous senescent female mice.

Our results also pointed to a heterogeneous cell population during pregnancy and postpartum at the symphyseal enthesis in primiparous and multiparous senescent animals. In primiparous females, from D19 to 3dpp, a similar fibroblast-like phenotype found in the BPR cell population, as described by Castelucci et al. [34], may indicate precursor cells that, although they have the same embryonic origin as the fibroblasts, can be differentiated according to the biomechanical properties, molecular and hormonal signaling, to which they are submitted [49]. Thus, this potential precursor cell population may be necessary for restoring cartilaginous tissues and their cell population at postpartum.

Although the differentiation into the BPR and the BDR was not evident in multiparous senescent animals, as observed in primiparous animals, most of the symphyseal enthesis cells presented the metabolically active phenotype. On the other hand, the symphyseal enthesis cells align parallelly to densely packed collagen fibers and present morphological characteristics of increased metabolic activity at D18. However, during postpartum, some cells with more condensed chromatin and few cytoplasmic organelles, probably quiescent cells with reduced metabolic activity, lead to a possibly reversible arrest in the cell cycle [50]. Notoriously, collagen fibrils become more compact and oriented in aged tissues [51].

The collagen fibers act as the main biomechanical stress reducer component, essential for stability through the forces exerted at the entheses [52]. The arrangement and the birefringence of collagen fibers showed a dynamism ranging from perpendicular to parallel and then returning to a perpendicular organization at the interpubic joint during pregnancy and postpartum in primiparous female mice. Given that the orientation of collagen fibers depends on the forces on the connective tissue [53,54,55], our data suggest that with the re-establishment of fibrocartilage, the biomechanics of the bone rapprochement stimulus modify the orientation of collagen fibers in response to shear forces in primiparous animals. In contrast, considering that multiparity and aging impact the postpartum return, causing the persistence of the IpL [33], the collagen fibers were compact and arranged parallelly to the opening of the joint on all days of our study. In addition, the helical organization of collagen fibers in crimps is an important factor for protection against rupture during the action of traction forces [56]. Due to the alignment of the cells with the collagen structure, the orientation of the cells reflects the orientation of the collagen fibers and thus the crimp morphology [57]. The crimps in the symphyseal enthesis of primiparous and multiparous senescent mice seem to be necessary to resist high stresses during pregnancy and postpartum. Our analysis suggests that, in senescent multiparous animals, the symphyseal enthesis is subject to constant stretching forces during and after pregnancy instead of shearing forces, possibly due to the persistent IpL.

Our work cannot distinguish whether a pure age or multiparous factor contributes to the observed changes. Our data corroborate the descriptions that the physiological processes impacted by parity and childbirth culminate in drastic adaptations of the birth canal and pelvic floor, as detailed by other authors [7,8]. However, the primiparous one-day postpartum on mouse pubic symphysis at 380 days old showed a shortened interpubic joint gap, which indicates that the aged mouse pubic symphysis can still be recovered after the first pregnancy [5]. Specifically, in primiparous female mice, the re-establishment of fibrocartilaginous tissue occurs due to the time-dependent expression of chondrocytic lineage markers [34], as temporal regulation of gene expression has been associated with the functional development of the enthesis [58].

The differential expression of Sox-9, Col2a1, and Runx-2 in elongated or rounded cells close to the BPR during pubic symphysis remodeling in primiparous animals may be indicative of the presence of a niche of osteochondral progenitor cells in a transient state, allowing commitment to one or more cell types according to molecular signaling [34]. However, due to parity and age, this niche of progenitor cells in the symphyseal enthesis is reduced or absent during pregnancy and postpartum in multiparous senescent animals. The transforming growth factor-β (TGF-β), bone morphogenetic protein (BMP), and Wnt signaling pathways are essential for the differentiation of chondrocytes during chondrogenesis in the embryonic development period of vertebrates. The inhibition of these pathways can lead to changes in the expression of osteochondral progenitor cell markers [59], such as Sox-9 and Runx-2 transcription factors [60,61].

*Sox-9* expression occurs throughout the chondrocyte differentiation process in developing cartilage from the embryonic period in vertebrates. It remains expressed in chondrocytes until hypertrophy in the growth plates and during adult life in articular cartilage [62], as well as the expression of *Runx-2*, which helps to regulate this differentiation by inducing chondrocyte hypertrophy during embryonic development in mice [63,64,65]. Furthermore, a coordinate transcriptional network for *Sox-9*, *Runx-2*, and *Col2a1* is described during chondroblast differentiation, specifically for cartilaginous tissues [66,67]. Specifically, in the primary cell culture of human articular cartilage, *Sox-9* has been shown to be responsible for maintaining chondrocyte phenotype and regulating *Col2a1* gene expression [68].

In primiparous animals, though *Sox-9* gene expression was upregulated, *Runx-2* and *Col2a1* gene expressions were downregulated until the end of pregnancy. However, these patterns are inverted at the interpubic joint in primiparous animals at postpartum. Thus, the *Sox-9*, *Runx-2*, and *Col2a1* gene expression profiles corroborate the morphological remodeling process of the pubic symphysis during the rapprochement of the pubic bones. In contrast, in multiparous senescent animals, although reduced levels of *Sox-9*, *Runx-2*, and *Col2a1* gene expression are seen throughout the sixth pregnancy and postpartum, there is a late expression of *Sox-9* in early postpartum and a *Col2a1* gene expression-distinct pattern during pregnancy and postpartum.

Our results demonstrate that the reduced levels of *Sox-9*, *Runx-2*, and *Col2a1* gene expression, the late expression of *Sox-9* in early postpartum, and the *Col2a1* gene expression-distinct pattern during pregnancy and postpartum may influence the maintenance of chondrocytic profile cells. *Runx-2* could suppress *Sox-9* during in vitro assays of human chondrocyte differentiation [69]. Furthermore, homozygous mutations in *Runx-2* have delayed ossification during embryonic development in mice [70,71]. Additionally, *Sox-9* can regulate genes, such as *Col2a1*, which is essential for chondroblast differentiation [66,67].

The spatiotemporal regulation of Sox-9 and Runx-2 protein expressions in primiparous female mice [34] is compatible with that observed during embryonic bone formation in mice [72,73,74]. However, in multiparous senescent mice, Runx-2 and Sox-9 showed a distinct pattern of immunostaining during the sixth pregnancy. Studies in transcriptome data with proteome data have identified an asymmetric distribution of enriched genes in enthesis and cartilage transcriptome, and Runx-2 and Sox-9 were identified as nodes within the protein–protein interaction network of transcription factors and growth factors enriched in the enthesis [75].

Many biological factors—not only these three, Col2a1, Runx-2, and Sox-9—may promote gradations in cell differentiation and subsequent symphyseal enthesis formation and remodeling. Keeping this in mind, other circulating factors and genes in regulating chondrogenesis and osteogenesis should be considered in further analysis, such as BMP-2, PTHrP, Ihh, scleraxis, BMP-12, and tenomodulin (for review, please see reference [35]). Importantly, osteoblast genesis processes should be further investigated considering new findings on the expression of osteocalcin, nerve growth factor, brain-derived neurotrophic factor, and different TRP channels involved in cellular Ca^2+^ and Mg^2+^ ion influx that regulates proliferation, differentiation, secretion, and apoptosis processes [76,77,78,79].

Despite this limitation, our findings indicate that the number of deliveries and age alter both gene and protein expression profiles of key molecules in the maintenance of the progenitor cell population of the chondrocytic and osteogenic lineages at the symphyseal enthesis in multiparous senescent animals, possibly compromising the re-establishment of fibrocartilage and hyaline cartilage cushions and impacting the recovery of mouse joint histoarchitecture, once Runx-2 and Sox-9 expressions are required for the development, homeostasis, remodeling and tissue repair of mouse fibrocartilaginous tendon entheses [35].

## 4. Materials and Methods

### 4.1. Animals

Young virgin female and male (three months old), as well as female and male retired breeder (twelve months old) C57BL/6J mice were obtained from the Multidisciplinary Center for Biological Investigation on Laboratory Animal Science (CEMIB) at the State University of Campinas (Unicamp). Retired breeders were selected from the reproductive matrix after having given five births. The young virgin female mice constituted the primiparous (P) group, and female retired breeders constituted the multiparous senescent (M) group. A period of 40 days without reproduction was kept for the multiparous senescent group to ensure that no remodeling occurred because of parturition. All mice were housed at 25 ± 2 °C under a 12 h light/dark cycle with free access to water and standard rodent chow. A breeding colony of mice was maintained at the Unicamp and used in the described studies. For timed pregnancies, breeding pairs were set up on Monday afternoon, and the next day between 7 and 9 a.m., a vaginal plug was considered to be an indicator of the first day of pregnancy (D1). PS or IpL were obtained from the following groups: day 12 of pregnancy (D12), D18, three days postpartum (3dpp), and 21dpp. A total of 72 females were used for light microscopy, immunohistochemistry, transmission electron microscopy, and real-time PCR. A total of 20 males were used for breeding. Between 11 a.m. and 12 p.m., the animals were anesthetized by an intraperitoneal injection of xylazine chloride (100–200 mg/kg) and ketamine (5–16 mg/kg) (Agribrands do Brasil, Paulinia, Brazil). After euthanasia by cervical dislocation, the medial portions of the pubic bones were removed. The animal experiments were conducted following the Guide for the Care and Use of Laboratory Animals, issued by the National Institutes of Health (NIH; Bethesda, MD, USA). All the experimental protocols were approved by the Institutional Committee for Ethics in Animal Experimentation (CEEA/IB/Unicamp), protocol 5491-1.

### 4.2. Light Microscopy

PS or IpL of primiparous and multiparous senescent female mice were fixed with 4% paraformaldehyde (Merck, Darmstadt, Germany) in 0.1 M phosphate-buffered saline (PBS, pH 7.4) for 48 h at 4 °C. Tissues were dehydrated in graded concentrations of alcohol and embedded in paraffin (Paraplast Kit embedding, Sigma, St. Louis, MO, USA) at 60 °C. PS or IpL paraffin-embedded 8μm sections from three different animals per time point were used for Masson Trichrome, Sirius Red and Hematoxylin staining, and immunohistochemistry. The sections were examined and imaged under light or polarized microscopy using an Eclipse E800 (Nikon, Tokyo, Japan) and a P6FL PRO digital camera (Optika, Ponteranica, Italy).

### 4.3. Transmission Electron Microscopy (TEM)

Samples of PS or IpL were fixed with 2.5% glutaraldehyde in 0.1 M sodium cacodylate buffer (pH 7.4) for 24 h at 4 °C, followed by post-fixation with 1% osmium tetroxide with 0.8% potassium ferrocyanide in 0.1 M sodium cacodylate buffer (pH 7.4) for 1 h at 4 °C. The samples were dehydrated in graded concentrations of alcohol and acetone, embedded in EMbed 812 Kit (Electron Microscope Science, Hatfield, PA, USA), and taken to polymerization for 72 h at 4 °C. Samples’ epoxy resin-embedded 70nm ultrafine sections from three different animals per time point were collected on copper grids, stained with uranyl acetate and lead citrate, and examined using a Tecnai G2 Spirit BioTWIN (FEI Company, Hillsboro, OR, USA). TEM was performed at the Electron Microscopy Laboratory of the Institute of Biology at Unicamp.

### 4.4. Immunohistochemistry

After paraffin removal, sections of PS or IpL were treated with 0.03% hydrogen peroxide for endogenous peroxidase inhibition, followed by blocking with 1% bovine serum bovine solution. The sections were incubated overnight at 4 °C with antibodies Sox-9 polyclonal (Invitrogen, Landsmeer, The Netherlands (PA5-81966), 1:100) and Runx-2 polyclonal (Invitrogen, Landsmeer, The Netherlands (PA5-82787), 1:100). As a negative control of the reaction, the primary antibody incubation step was omitted. On the next day, samples were incubated with the EnVision Plus System (Dako, Carpinteria, CA, USA, (K4061)) for 1 h at room temperature. After the washing step, incubation with the peroxidase developer solution was carried out for 3 min with a substrate mixture of 0.5 mg/mL 3.3-diaminobenzidine (Dako, Carpinteria, CA, USA, (K3468)). The section was then counterstained with Harris hematoxylin. The sections of enthesis were imaged using an Eclipse E800 light microscope (Nikon, Tokyo, Japan) and a P6FL PRO digital camera (Optika, Ponteranica, Italy).

The immunopositive cells for Sox-9 and Runx-2 were identified based on their immunoreactivity. Cells at hyaline cartilage and fibrocartilage were counted at D12 and 21dpp in primiparous group. In the other groups, cells at symphyseal enthesis were counted (demarcated areas in the Figure 4 and Figure 5). Three random images were analyzed from each animal for each day of pregnancy and postpartum by QuPath 0.4.2 [80].

### 4.5. Real-Time PCR (qPCR)

The interpubic tissues from three (D18 and 3dpp) or six (D12 and 21dapp) different animals per time point were used for qPCR. Briefly, the RNA was extracted using TRIzol Reagent (Invitrogen, Carlsbad, CA, USA) and cDNA using RevertAid H Minus First Strand cDNA Synthesis Kit (Thermoscientific, Vilnius, Lithuania) to obtain 200 ng of cDNA for each day of study. Both were carried out according to the manufacturer’s recommendations. Real-time PCR was performed using SYBER Green (Applied Biosystems, Foster City, CA, USA) in an Applied Biosystems StepOne Plus (Walthan, MA, USA) at the Genomics and Bioenergy Laboratory of the Institute of Biology at Unicamp. Each gene was normalized to the expression of the housekeeping gene *36b4*, officially known as ribosomal protein, large, P0 (*Rplp0*). To qualify and determine the fold increase in gene expression, a 2^−ΔΔCt^ mathematical model was utilized, and the results were normalized with the D12 group. All reactions were performed in triplicate on the same plate. The primers for *36b4*, *Sox-9*, *Runx-2*, and *Col2a1* were purchased from IDT (Integrated DNA Technologies, Coralville, IA, USA) (see Appendix A for primer sequences).

### 4.6. Statistical Analysis

The relative gene expression of qPCR analysis was performed using two-way ANOVA followed by the Bonferroni test, with *p*-value < 0.05 indicating significance when performed with GraphPad Prism 8.0.2 (GraphPad Software, La Jolla, CA, USA). Data were compared over the first (P vs. P) or sixth (M vs. M) pregnancy or between the first and sixth (P vs. M) pregnancy on the same day of study. All data are presented in graphs as the mean value ± standard error (SE).

## 5. Conclusions

Morphological and molecular differences were found at the symphyseal enthesis in multiparous senescent animals compared to primiparous. Our study allows us to infer that, in the interpubic joint, the enthesis appears to be a dynamic transition, which develops during pregnancy and regresses at postpartum in primiparous animals. Meanwhile, gene expression and immunohistochemistry showed the spatiotemporal regulation of chondrogenic and osteogenic markers such as Sox-9, Runx-2, and Col2a1 during the formation of the symphyseal enthesis and the IpL during pregnancy and postpartum in primiparous females. In contrast, multiparous senescent females showed a symphyseal enthesis structure throughout pregnancy and postpartum without clear BPR and BDR delimitations on our study days. The symphyseal enthesis cells are metabolically active during pregnancy and postpartum, which may indicate responsiveness to pregnancy stimuli. However, they have a reduced expression of chondrogenic and osteogenic markers and are immersed in densely packed collagen fibers contiguous to persistent IpL. These findings may indicate that the number of deliveries and the age alter both gene and protein expression profiles of key molecules in the maintenance of the progenitor cell population of the chondrocytic and osteogenic lineages at the symphyseal enthesis in multiparous senescent animals, possibly compromising the re-establishment of fibrocartilage and hyaline cartilage cushions and impacting the recovery of mouse interpubic joint histoarchitecture. Then, our findings shed light on ligament remodeling biology and its contribution to connective tissue disorders in women, such as PSD—although a rare condition causes serious pain during the postpartum period, and POP—a prevalent disease associated with the number of births and age. Both conditions have implications in orthopedic and urogynecological practice.

## Figures and Tables

**Figure 1 ijms-24-04573-f001:**
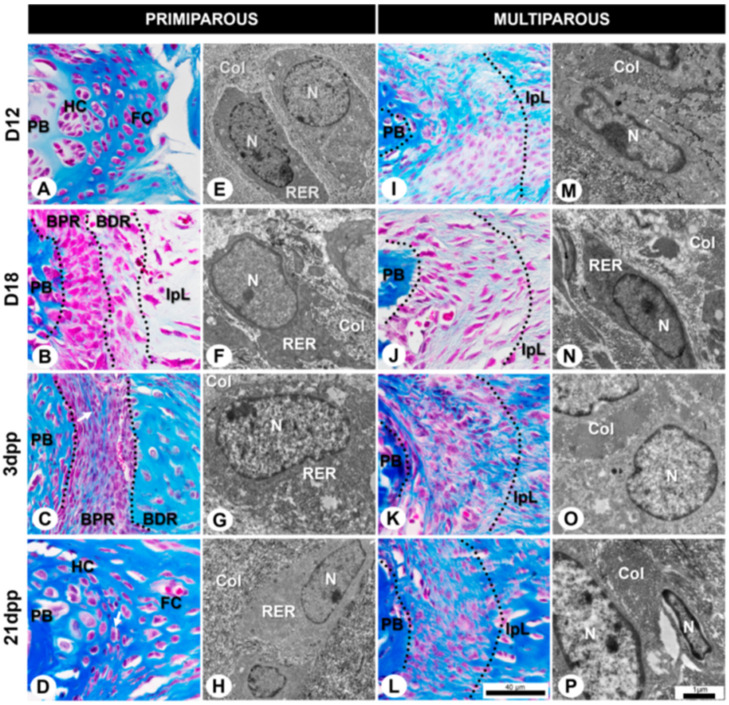
Representative cells and extracellular matrix (ECM) morphological aspects of symphyseal enthesis during pregnancy and postpartum in primiparous (**A**–**H**) and multiparous senescent (**I**-**P**) female mice at light and transmission electron microscopy. (**A**) At D12, pubic symphysis (PS) architecture is composed of the fibrocartilaginous disc (FC) between hyaline cartilage (HC). (**B**,**C**) At D18 and 3dpp, between pubic bones (PB) and interpubic ligament (IpL), see demarcated areas with dotted lines that approximately correspond to two distinct regions in the symphyseal enthesis: bone proximal region (BPR) with numerous oval cells next to each other and bone distal region (BDR) with elongated and sparse cells. (**D**) At 21dpp, the BPR and BDR of the enthesis are indistinguishable, and a similar non-pregnant joint morphology is restored. See the organization of cells around the PB and the isogenous groups (arrow). (**I**–**L**) Symphyseal enthesis of multiparous senescent mice with persistent IpL during pregnancy and postpartum. BDR and BPR are almost indistinguishable in demarcated areas with dotted lines. (**E**–**H**,**M**–**P**) Ultrastructure of symphyseal enthesis cell types and ECM aspects during pregnancy and postpartum in primiparous and multiparous senescent female mice. Observe the presence of a developed rough endoplasmic reticulum (RER) in the cytoplasm, nucleus (N), and collagen fiber (Col) arrangement. (**A**–**D**,**I**–**L**) Masson trichrome staining, scale bar = 40 μm. (**E**–**H**,**M**–**P**) Transmission electron microscopy, scale bar = 1 μm.

**Figure 2 ijms-24-04573-f002:**
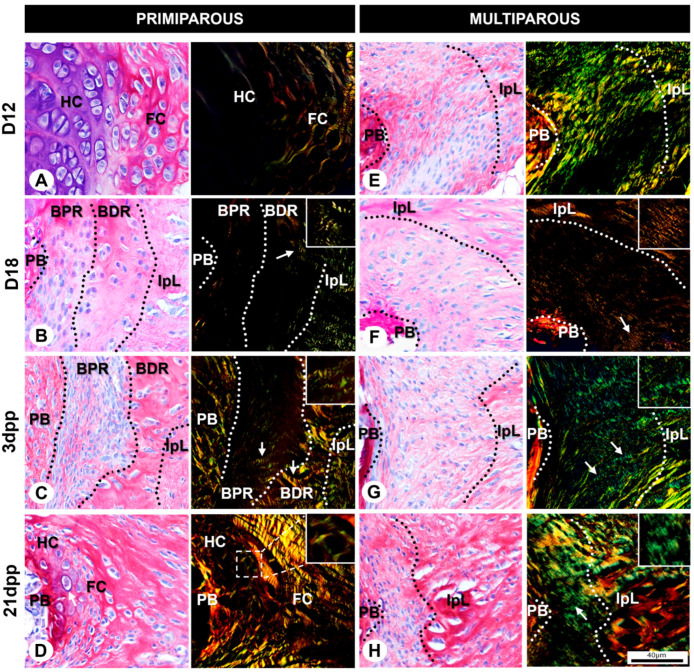
Arrangement of collagen fibers at conventional and polarized light microscopy of symphyseal enthesis in transverse sections during pregnancy and postpartum in primiparous (**A**–**D**) and multiparous senescent (**E**–**H**) female mice. (**A**) At D12, observe the transition of the fibrocartilaginous disc (FC) to hyaline cartilage (HC) at the pubic symphysis (PS) architecture. Specifically in the FC, observe the collagen fiber at a perpendicular orientation related to the PS. (**B**,**C**) By interpubic ligament (IpL) formation, note collagen fiber crimps at the bone proximal region (BPR) and bone distal region (BDR) (arrows and insets). (**D**) At 21dpp, at the restored HC, see the birefringence around cells in their lacunae with a noticeable basophilic territorial matrix (arrow) next to the pubic bone (PB). (**E**–**H**) Compare the arrangement of collagen fibers at the symphyseal enthesis in the transverse section during pregnancy and postpartum in the multiparous senescent female mice. Note the parallel organization of the birefringent collagen fibers and the presence of crimps (arrows and insets). Sirius Red and hematoxylin staining, scale bar = 40 μm.

**Figure 3 ijms-24-04573-f003:**
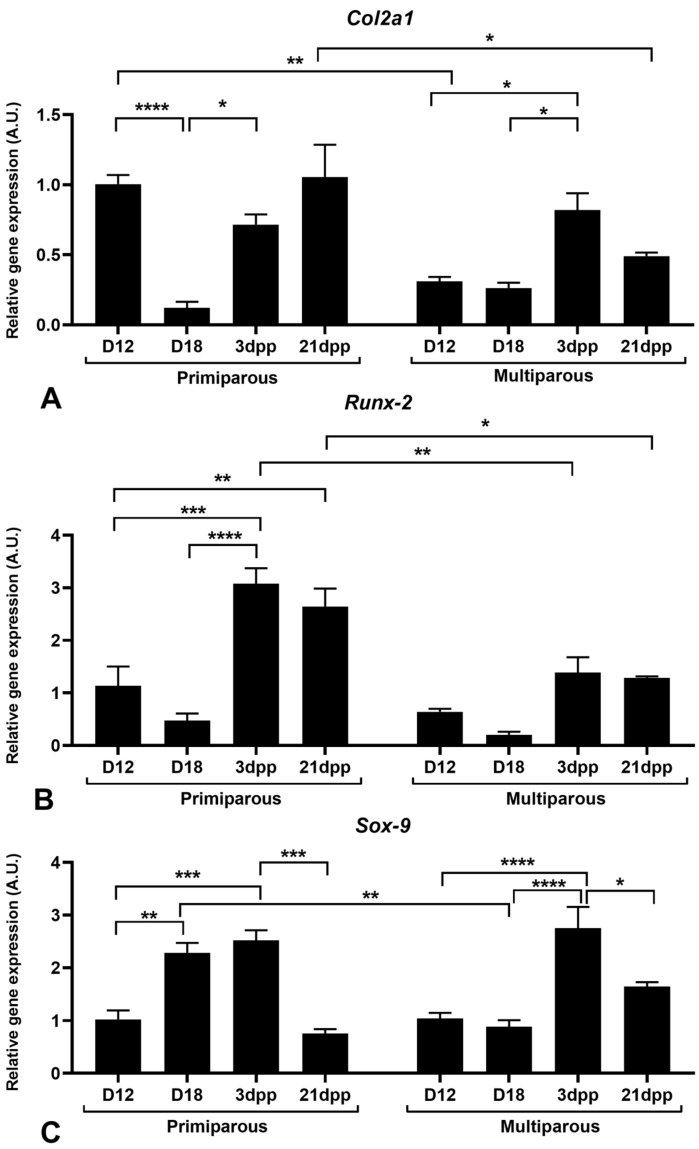
Spatiotemporal gene expression of *Col2a1* (**A**), *Runx-2* (**B**), and *Sox-9* (**C**) at the interpubic joint during pregnancy and postpartum remodeling in primiparous and multiparous senescent female mice by qPCR. Two-way ANOVA with Bonferroni test. Significance *p*-value < 0.05 (*), < 0.01 (**), < 0.001 (***), < 0.0001 (****). A.U.= arbitrary unit. N = 3 animals per group (D18, and 3dpp); 6 animals per group (D12, and 21dpp).

**Figure 4 ijms-24-04573-f004:**
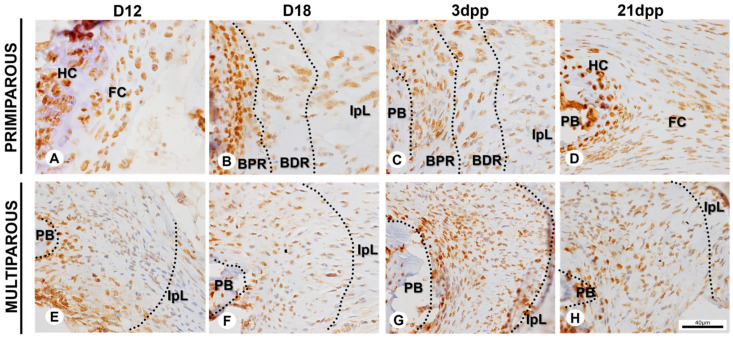
Immunopositive cells for Runx-2 at the symphyseal enthesis during pregnancy and postpartum in primiparous and multiparous senescent female mice. (**A**–**D**) Runx-2 immunostaining at hyaline cartilage (HC), fibrocartilage (FC), bone proximal region (BPR), bone distal region (BDR), and interpubic ligament (IpL). (**E**–**H**) Runx-2 immunostaining at the symphyseal enthesis and IpL. If distinguishable, demarcated areas with dotted lines correspond approximately to BPR and BDR. Immunohistochemistry (IHC) and hematoxylin staining. Scale bar = 40 μm.

**Figure 5 ijms-24-04573-f005:**
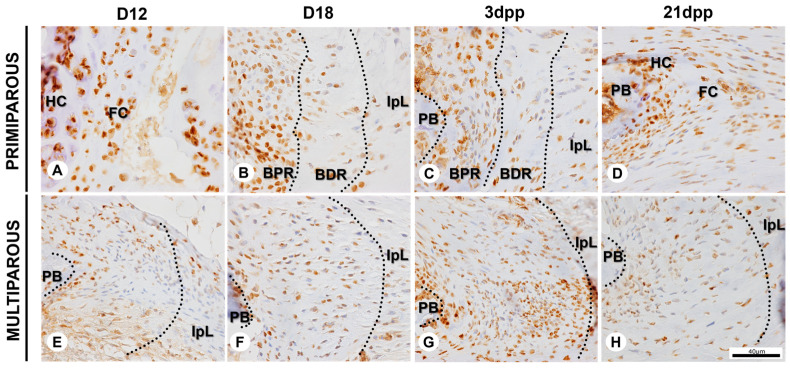
Immunopositive cells for Sox-9 at the symphyseal enthesis during pregnancy and postpartum in primiparous and multiparous senescent female mice. (**A**–**D**) Sox-9 immunostaining at hyaline cartilage (HC), fibrocartilage (FC), bone proximal region (BPR), bone distal region (BDR), and interpubic ligament (IpL). (**E**–**H**) Sox-9 immunostaining at the symphyseal enthesis and IpL. If distinguishable, demarcated areas with dotted lines correspond approximately to BPR and BDR. Immunohistochemistry (IHC) and hematoxylin staining. Scale bar = 40 μm.

## Data Availability

Not applicable.

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
