# Peer review of "Multiparity and Aging Impact Chondrogenic and Osteogenic Potential at Symphyseal Enthesis: New Insights into Interpubic Joint Remodeling"

_ijms, 2023, doi:10.3390/ijms24054573_

Round 1

Reviewer 1 Report

The authors in their work aimed to understand the effects of parity and age on chondrogenesis and osteogenesis at the symphyseal enthesis in primiparous mice and multiparous senescent mice during pregnancy and postpartum and the biological factors that are necessary for the development, homeostasis, and remodeling of mouse entheses, such as the expression of Runx-2, Sox-9, and collagen type II  (Col2a1) genes. 

The biological investigation was limited to these genes and factors but the mechanism can be more complex than expected involving other genes and factors.

Mayor concern :

In the discussion sections, the authors should discuss and consider the possible role of other circulating factors and genes in the regulation of chondrogenesis and osteogenesis such as the Nerve Growth Factor, Brain-Derived Neurotrophic Factor, and Osteocalcin Gene and their receptor signaling, and the TRPV1 regulating intracellular calcium and osteoblast genesis (see below).

Oxytocin/Osteocalcin/IL-6 and NGF/BDNF mRNA Levels in Response to Cold Stress Challenge in Mice: Possible Oxytonic Brain-Bone-Muscle-Interaction.

Camerino C, et al., Front Physiol. 2019 Nov 27;10:1437. doi: 10.3389/fphys.2019.01437. eCollection 2019.

Zoledronic Acid Modulation of TRPV1 Channel Currents in Osteoblast Cell Line and Native Rat and Mouse Bone Marrow-Derived Osteoblasts: Cell Proliferation and Mineralization Effect.

Scala R, et al., .Cancers (Basel). 2019 Feb 11;11(2):206. doi: 10.3390/cancers11020206.

Nerve Growth Factor, Brain-Derived Neurotrophic Factor and Osteocalcin Gene Relationship in Energy Regulation, Bone Homeostasis and Reproductive Organs Analyzed by mRNA Quantitative Evaluation and Linear Correlation Analysis.

Camerino C, et al., .Front Physiol. 2016 Oct 13;7:456. doi: 10.3389/fphys.2016.00456. eCollection 2016.

Reviewer 2 Report

The authors investigated the effects of parity and age on chondrogenesis and osteogenesis at the symphyseal enthesis in primiparous mice and multiparous senescent mice during pregnancy and postpartum. The authors found that the number of deliveries and age might alter both gene and protein expression profiles of key molecules in the maintenance of the progenitor cell population of the chondrocytic and osteogenic lineages at the symphyseal enthesis in multiparous senescent animals, possibly compromising the re-establishment of fibrocartilage and hyaline cartilage cushions impacting the recovery of mouse joint histoarchitecture.

Comments:

The reviewer has some concerns as follows:

1. In the Methods section, the mating conditions of the animals should be clearly described. Moreover, the information for male mice is lacking that should be confirmed if it has been approved by the Institutional Committee for Ethics in Animal Experimentation.

2. The low sample size is one of the major concerns. In the Methods section, the authors described “Three animals were used for each experimental group for light microscopy, transmission electron microscopy, immunohistochemistry, and real-time PCR…”.

3. Another major concern is that the age-matched control mice for multiparous senescent mice are lacking. Lacking this age-matched control group, it cannot be distinguished whether a purely geriatric factor or multiparous factor contributes to the observed changes.

4. In Figure 3, the data presentation and statistical analysis are confusing. There are too many symbols and numbers in the figures and the data among groups are not easy to compare. What is the significance of these prenatal and postnatal changes for these gene expression levels?

5. The IHC data shown in Figures 4 and 5 are not convincing. The changes are not easy to read. The quantification can be shown.

Reviewer 3 Report

This is a report of a very ambitious and original study that investigated the effects of multiple births and aging on the epiphyseal chondrogenic and osteogenic potential of pubic symphysis. While it is a very interesting paper and has some useful results, there are a few points that seem inadequate. I think that the following points require mention or correction.

#1. it is unclear whether the findings described in the Introduction are those observed in humans, in laboratory animals such as mice, or in both. The Discussion should clearly distinguish between the two and discuss whether the phenotypes that support the results of this study, especially in mice used in this experiment, are consistent with those observed in actual mouse births.

#2. The current experimental details cause ambiguous interpretations of the results as to whether the effects on epiphyseal chondrogenic and osteogenic potential are due to multiple births, aging, or a combination of the two. For example, would it be possible to verify whether similar results would be obtained if physical traction were applied to the pubic symphysis experimentally at matched ages? Or what about using mice that are older but have never given birth as controls?

Round 2

Reviewer 1 Report

The manuscript has been improved in the current version

Reviewer 2 Report

This revised manuscript can be accepted. No further comments.

Reviewer 3 Report

The authors have revised their original manuscript according to the reviewers’ comments. I think that this revised manuscript is better organized and suitable for publication.
